# KD Diagnosis Does Not Increase Cardiovascular Risk in Children According to Dynamic Intima–Media Roughness Measurements

**DOI:** 10.3390/jcm11051177

**Published:** 2022-02-22

**Authors:** Miriam König, Theresa Ullmann, Belén Pastor-Villaescusa, Robert Dalla-Pozza, Sarah Bohlig, Arno Schmidt-Trucksäss, Joseph Pattathu, Nikolaus A. Haas, André Jakob

**Affiliations:** 1Department of Paediatric Cardiology, Ludwig-Maximilians-University of Munich, 81377 Munich, Germany; miriam.kab@t-online.de (M.K.); belen.pastor@imibic.org (B.P.-V.); robert.dallapozza@med.uni-muenchen.de (R.D.-P.); sarah.wala93@gmail.com (S.B.); joseph.pattathu@med.uni-muenchen.de (J.P.); nikolaus.haas@med.uni-muenchen.de (N.A.H.); 2Institute for Medical Information Processing, Biometry and Epidemiology, Ludwig-Maximilians-University of Munich, 81377 Munich, Germany; tullmann@ibe.med.uni-muenchen.de; 3Department of Sport, Exercise and Health, Division Sport and Exercise Medicine, University of Basel, 4052 Basel, Switzerland; arno.schmidt-trucksaess@unibas.ch

**Keywords:** Kawasaki disease, intima–media roughness, vascular disease

## Abstract

Background: Kawasaki Disease (KD) is a generalized vasculitis in childhood with possible long-term impact on cardiovascular health besides the presence of coronary artery lesions. Standard vascular parameters such as carotid intima–media thickness (cIMT) have not been established as reliable markers of vascular anomalies after KD. The carotid intima–media roughness (cIMR) representing carotid intimal surface structure is considered a promising surrogate marker for predicting cardiovascular risk even beyond cIMT. We therefore measured cIMR in patients with a history of KD in comparison to healthy controls to investigate whether KD itself and/or KD key clinical aspects are associated with cIMR alterations in the long-term. Methods: We assessed cIMR in this case-control study (44 KD, mean age in years (SD); 13.4 (7.5); 36 controls, mean age 12.1 (5.3)) approximately matched by sex and age. Different clinical outcomes such as the coronary artery status and acute phase inflammation data were analyzed in association with cIMR values. Results: When comparing all patients with KD to healthy controls, we detected no significant difference in cIMR. None of the clinical parameters indicating the disease severity, such as the persistence of coronary artery aneurysm, were significantly associated with our cIMR values. However, according to our marginally significant findings (*p* = 0.044), we postulate that the end-diastolic cIMR may be rougher than the end-systolic values in KD patients. Conclusions: We detected no significant differences in cIMR between KD patients and controls that could confirm any evidence that KD predisposes patients to a subsequent general arteriopathy. Our results, however, need to be interpreted in the light of the low number of study participants.

## 1. Introduction

Kawasaki Disease (KD) is a generalized vasculitis in children, still of unknown etiology [1]. Other than boys < 5 years being most often affected ^1^, its incidence also reveals a geographic and ethnic susceptibility, as children of Asian origin are affected more often than those of Caucasian origin: the highest incidence in children aged 0–4 years is in Japan, with 359/100,000 in 2018 [2], whereas the annual incidence in Germany is 7.2/100,000 [3]. Clinically, KD presents with persistent fever, cervical lymphadenopathy, non-exudative conjunctivitis, enanthema, polymorphous exanthema and reddened palms and soles, with subsequent scaling on the toes and fingers. Coronary artery aneurysms (CAA) are KD’s most feared complication, with a strong impact on long-term morbidity; they still occur in approximately 5% of all children who undergo intravenous immunoglobulin (IVIG) treatment [1,4], and in even 25% of untreated children [4]. Its acute clinical presentation already shows that the inflammatory process is not only limited to the coronary arteries, but is probably attributable to vascular changes in various tissues and organs, e.g., the gastrointestinal tract (pain, vomiting, diarrhea), meninges (aseptic meningitis) or lung (interstitial pneumonitis) [5]. In line with these manifestations, a pathological serum lipid profile [6], prolonged endothelial dysfunction [7,8] and persistent endothelial injury, identifiable as increased levels of circulatory endothelial cells and endothelial microparticles [9], are observed as KD’s long-term effects.

Vascular inflammation attributable to KD can also influence the carotid’s intimal and media layers. In patients with CAA, carotid intima–media thickness (cIMT)—measured as the distance between the lumen–intima interface and media–adventitia boundary of the common carotid artery—was found to be thicker than in KD patients without coronary artery lesions [10]; however, the data on cIMT in patients with a KD history are inconsistent [8,9]. Carotid intima–media roughness (cIMR) describes the common carotid artery’s surface texture. It evaluates the surfaces of the carotid artery’s intima and media layer, even beyond the intima–media thickness, and allows us to detect signs of arteriopathy, such as plaque formation, as irregularities in the intima–media layers [11]. Persisting irregularities affecting the vessels’ inner wall after an inflammatory process such as KD may be diagnosed more accurately than by altered cIMT. Furthermore, cIMR has two advantages over cIMT: cIMR does not seem to be prone to age-related physiological vascular changes and shows a closer correlation with pathological changes and the atherosclerotic process than cIMT [11,12]. We therefore analyzed cIMR in a case-control study-design to investigate whether KD itself and/or clinical key aspects of KD are associated with cIMR alterations and therefore indicate a possible general arterial vasculopathy in the long term.

## 2. Materials and Methods

### 2.1. Study Population

Patients with KD were recruited and examined at the paediatric cardiology department at Ludwig Maximilian University of Munich. All patients’ KD diagnoses had been verified following the American Heart Association (AHA) guidelines. The Z-score of each coronary artery was calculated according to Dallaire et al. [13], with a Z-score > 2.5 defining a CAA.

Healthy controls were recruited among the patients′ personal contacts. The patients or parents were asked to present in a fasting state and bring along a sex- and age-identical friend or sibling to the appointment. We took a detailed medical history and carried out physical examinations, ECGs, and echocardiography in all patients and controls. The following applies to the healthy control children: absence of any cardiac, infectious, oncologic, rheumatic, neurologic or endocrine disease (e.g., diabetes mellitus) to eliminate any impact on the cardiovascular system. Further inclusion criteria were: normal resting ECG (12 leads), body mass index (BMI) < 30 kg/m^2^, blood pressure < 97 th percentile and normal echocardiography.

After at least 10 min in resting state, blood tests and blood pressure were taken and the cIMR ultrasound examination was performed by a pediatric cardiologist, experienced in cIMT measurement.

### 2.2. Ultrasound Examination

CIMR was determined via an ECG-triggered sonographic examination. An ultrasound device from Philips (IE33) equipped with a linear transducer (L11-3) was used for the measurement. All examinations were carried out according to a standardized protocol for right and left common carotid artery. In a supine position, patients turned their head 30 degrees to the opposite side to create optimal visual conditions. The common carotid artery (CCA) was visualized in the longitudinal plane, and the intima–media layer was visible as typical double lines. Two cine-loops in B-mode were recorded on each side, proximal to the carotid bulb. The first possible 10 mm section proximal to the carotid bulb of sufficient image quality was selected for analysis. To measure the CCA’s far wall, we relied on the end-diastolic and end-systolic phase of the heart cycle, defined via the top of the QRS-complex and the end of the T-wave, in order to minimize intrabeat variability. Dynamic Artery Analysis (DYARA) software was used to automatically identify the IM layers. This software has been described and validated in adult and pediatric cohorts [11,14,15,16]. To determine the media–adventitia border and lumen–intima border of the far wall and the media–adventitia interface of the near wall, DYARA software analyses a 10-mm segment (266 pixels) (see Figure 1).

The regression line of all cIMT values within the 10-mm segment was calculated, and the respective regression line’s deviation was determined. CIMR was calculated as the arithmetic mean of the deviation from the regression line, according to the ISO definition (International Organization for Standardization, Geneva, Switzerland). The higher this deviation, the greater the roughness [11] (Figure 2).

### 2.3. Statistical Analysis

Statistical analysis was performed with the program R (Version R-4.0.2). Data are indicated as the mean and standard deviation (SD) unless otherwise specified. To check the homogeneity of sex distribution between patients with KD and controls, the χ2 test was applied. For the participants’ clinical characteristics (age, body mass index (BMI), blood pressure and blood lipids determined on the day of presentation) and cIMR measurements (end-diastolic and end-systolic), we assessed differences at baseline per groups via Welch′s *t*-test (an adaptation of the standard Student’s *t*-test for unequal variances). Furthermore, to evaluate differences between end-diastolic and end-systolic cIMR measurements, the paired-samples *t*-test was applied. For correlations between clinical parameters (BMI, blood pressure and lipid profile) and cIMR, Pearson’s coefficient was assessed. The correlation analyses were conducted for all study participants together as well as stratified by KD and control patients.

Multiple linear regression analysis adjusted for age, sex, weight and height was performed to study the association between KD (absence of disease (control group) = reference) and cIMR measurements. In relation to the coronary artery lesion, patients were subdivided into the following three groups: A: never had CAA, B: CAA only during acute phase of KD and C: persistent CAA. In one regression model, each group (A, B and C) was compared to the healthy control group (i.e., group membership was encoded as a four-level categorical predictor with the healthy control group as baseline). In another regression model, we compared group A to group B + C (without including the healthy control group).

In the second step, we evaluated the associations between clinical KD acute phase parameters and the end-diastolic and end-systolic cIMR. The following acute phase variables were used as predictors in a regression model (with either end-diastolic or end-systolic cIMR as the dependent variable): aneurysm status (groups B + C vs. group A), IVIG non-responder, duration of fever (days), C-reactive protein (CRP) levels, thrombocytes, leucocytes and hemoglobin. As before, we adjusted for age, sex, weight and height (measured at the time of our study).

In all analyses, a *p*-value of < 0.05 was considered significant.

## 3. Results

### 3.1. Comparison of Baseline Characteristics between Patients and Controls

A total of 44 patients with KD (15 female and 29 male, mean age (SD) in years: 13.4 (7.5)) and 36 healthy controls (14 female and 22 male, mean age in years: 12.1 (5.3)) were included in this investigation. One patient (out of the original 45 patients) had to be excluded from the study due to poor image quality, as intima–media boundaries were not precise enough for analysis.

The examination was carried out, on average, 8.1 years after the acute phase of KD (range: 7 months to 26.9 years). Patients with KD and controls did not differ significantly in age, sex or BMI (mean BMI (SD) in kg/m^2^: KD 18.2 (3.7) vs. controls 18.1 (3.1). Furthermore, our assessments of blood pressure hemodynamics (e.g., mean blood pressure (SD) in mmHg KD 117/71 (13.6/10.0) vs. controls 113/68 (8.2/9.2)) and blood–lipid values (e.g., mean LDL-cholesterol (SD) in mg/dL: KD 104.7 (25.5) vs. controls 104(26.2)) revealed no significant differences between the two groups. A detailed comparison of baseline characteristics in patients with KD and controls is provided in Table 1.

### 3.2. Carotid Intima–Media Roughness

Two loops per carotid artery side were analyzed and used to calculate mean cIMR values. Mean cIMR was calculated as the mean of all four recorded loops of both carotid arteries. For one patient, only one loop per side (left/right) could be recorded instead of two; the mean cIMR values were calculated from the available measurements. For one control person, one of the two end-diastolic cIMR values recorded on the right carotid artery had to be removed due to a measurement error. For this control person, the mean end-diastolic cIMR was thus calculated from two loops on the left and one on the right side. We also calculated end-systolic and end-diastolic cIMR values in each study participant.

No statistically significant difference appeared after comparing right-sided and left-sided cIMR values from all study participants together (these data are not presented here, but are available upon request). However, we did observe that the mean end-diastolic cIMRs were significantly higher than the end-systolic values when taking all study participants together (N = 80, mean end-diastolic cIMR = 0.041 mm vs. mean end-systolic cIMR = 0.039 mm, 95% CI = (0.00005; 0.0036) (CI of the difference between end-diastolic and end-systolic cIMR), *p*-value = 0.044).

BMI, blood pressure and blood lipid values mostly did not correlate significantly with our cIMR measurements, except for VLDL (mg/dL), which correlated significantly with end-diastolic cIMR in the healthy control group (r = −0.502, *p*-value = 0.012) and when taking all study participants together (r = −0.334, *p*-value = 0.026). However, due to the high number of tests performed overall, this result should be regarded with caution (we did not correct for multiple testing). In the patient group, VLDL (mg/dL) did not correlate significantly with end-diastolic cIMR. The full correlation results are available upon request.

Differences in heart cycle-specific cIMR between patients and controls were analyzed first via the Welch’s unequal variances *t*-test. Neither the end-diastolic cIMR (patients 0.040 mm vs. controls 0.042 mm (95% CI = (−0.0059; 0.0034) (CI of the difference between cIMR in patients and controls); *p*-value = 0.590) nor end-systolic cIMR (0.039 mm vs. 0.039 mm (95% CI = (−0.0043; 0.0037)) (CI of the difference between cIMR in patients and controls); *p*-value = 0.886) differed significantly between groups (see also Figure 3).

To further evaluate KD’s influence on cIMR, we performed linear regression analysis adjusted for age, sex, weight and height, but again, cIMR values revealed no significant association with KD (see Table 2).

Comparing end-diastolic cIMR measurements to age and sex-specific normative data on children aged 8 to 18 years [17], we identified two patients with KD (belonging to Group A) and one control exceeding the 97th percentile.

### 3.3. Influence of Coronary Artery Status and Acute Phase Parameters on cIMR

The age of KD patients at the acute phase of the disease ranges from 2 months to 11.7 years. Twelve patients developed CAA during the acute phase of KD, six of whom had recovered to normal dimensions in the follow-up echocardiographic analysis. Regarding laboratory results, for each patient, the most pathological value measured during the acute phase of KD was considered for our analysis. For further acute-phase parameters, see Table 3.

We also applied multivariate regression analysis to investigate any associations between the coronary artery status, acute phase parameters and cIMR. In relation to the CAA, patients with KD were divided into three subgroups (A: never had CAA N = 27, B: regressed CAA N = 6, C: persisting CAA N = 6, missing values: N = 5). Due to the five persons with unknown CAA status, the number of KD patients in the analyses was reduced to 39 persons. Patients with KD with evidence of coronary artery involvement (Group B and C) had slightly higher mean end-systolic cIMR values than patients in Group A (mean end-systolic cIMR: Group A 0.038 mm, Group B 0.043 mm, Group C 0.041 mm). For end-diastolic cIMR values, there was no clear tendency (mean end-diastolic cIMR: Group A 0.040 mm, Group B 0.038 mm, Group C 0.042 mm). However, according to the multiple linear regression analyses, the group differences were not statistically significant for either end-systolic or end-diastolic values (see Table 4).

Multiple linear regressions showed that acute phase parameters such as being a non-responder to intravenous immunoglobulins (IVIG), duration of fever (days), CRP levels, thrombocytes, leucocytes and hemoglobin were significantly associated with neither the end-systolic nor the end-diastolic cIMR (these data are not presented here but are available upon request). These regression analyses were limited to 29 patients with complete values for all the variables.

## 4. Discussion

### 4.1. Dynamic Carotid Intima–Media Roughness, a Potential Vascular Parameter

In this study, we investigated cIMR as a potential surrogate marker for long-term vascular anomalies in children with a history of KD. To the best of our knowledge, the cIMR in children with KD has not been investigated until now. cIMR might reveal an intima–media anomaly more thoroughly than the thickness alone. However, we found that our KD patients’ cIMR—including both the systolic and diastolic values— failed to differ significantly from our healthy controls’ cIMR. Furthermore, 95.6% of all cIMR values of the KD patients concurred with published cIMR normative data for children [17]; they were only exceeded in both groups by a total of three children. Moreover, those patients with KD and a more severe clinical presentation (represented, for example, by CRP levels (mean CRP levels 115.4 mg/L), being refractory to IVIG, or developing CAA), did not demonstrate a “rougher” intima–media surface.

Following the characteristics of cIMT, we found that cIMR measured during the heart cycle’s diastole was significantly higher than end-systolic cIMR. As assumed for cIMT, the most probable reason for this is the lumen diameter’s physiological expansion during systole, causing the artery’s intima–media complex to become thinner. A study from 2017 even implicates that this effect is clearly stronger in childhood than in adulthood, as they reported a cIMT increase by 8.8% (38 µm) in four-year-olds compared to only 3.8% (18 µm) in adults [18].

AHA guidelines recommend the acquisition of end-diastolic cIMT values for cardiovascular risk assessment [19]. The inconsistency of published normative cIMT values regarding heart cycle specific measurement can lead to misinterpretation, e.g., the ARIC study [20] where cIMT was acquired in peak systole vs. the cardiovascular health study [21] where the phase of cardiac cycle was not controlled. This pitfall should be taken into account in future studies investigating cIMR to ensure consistent investigation procedures, especially concerning pediatric populations of very different ages.

cIMT studies in adults showed that atherosclerotic anomalies such as plaque formation or a granulated intima–media layer are linked to cardiovascular risk factors and transient ischemic lesions or stroke, and may be a major criterion for cardiovascular events [22,23,24,25]. cIMR describes the amount of variation in a set of cIMTs and is supposed to quantify irregularities in the IM layers caused by such plaque formations or granulations as a sign of arterial injury. In adults, cIMR is known to correlate closely with arterial hypertension and coronary artery disease [11,26]. Wu et al. found cIMR was significantly associated with the Framingham heart score, and that it increased in conjunction with the number of cardiovascular risk factors. Smoking status, systolic blood pressure and hyperlipidemia, especially, proved to be independently associated with cIMR [27]. These results reveal that cIMR is a new parameter facilitating the assessment of arteriopathy and cardiovascular risk. However, in our pediatric study participants, we failed to detect any significant association between cIMR values and blood lipids and/or blood pressure, except for a negative correlation of cIMR to VLDL, acknowledging that all values had been within the normal range. Yet, up to now, only few studies assessed the cIMR in children. Nevertheless, significant positive correlation between cIMR and cardiovascular risk factors such as body mass index and a strong negative correlation to physical fitness, measured as maximum oxygen uptake, have been found ^17^. In one case report, the cIMR in a 12-year-old boy with type 1 diabetes diminished during consequent blood glucose control from 0.048 mm to 0.036 mm (initial HbA1 c level: 15%; after 41 months: 8.2%) [28]. Patients with diabetes are known to carry an increased cardiovascular risk, but that risk decreases as HbA1 C-levels return to normal [29]. cIMR might therefore be a parameter to reveal cardiovascular risk not only in adults, but also in children.

Apart from its effect on the coronary arteries, the evidence of KD’s long-term effect on the vascular system is inconsistent. Dalla Pozza et al. [10] detected significantly higher cIMT values in children after a KD diagnosis than in healthy controls and normative data on Western children. Furthermore, children with persistent CAA even demonstrated an additional significant increase in cIMT. Another investigation, including 92 patients with KD, found no difference in cIMT and arterial stiffness, measured as pulse wave velocity (PWV). However, the same study reported that different markers of endothelial injury—such as circulating endothelial cells (CECs), endothelial microparticles (EMPs) and soluble adhesion molecules—were significantly higher in patients with KD than in healthy controls; they were highest in patients with persistent CAA [9]. A systematic review evaluated 30 studies investigating endothelial dysfunction, arterial stiffness and cIMT in patients with a KD history and found that, while arterial stiffness and endothelial dysfunction seem to be abnormal, especially in CAA-positive patients with KD, studies measuring cIMT were inconsistent [8]. cIMR might give us information on intima–media anomalies even beyond the thickness itself. However, the results we obtained from our patient cohort did not indicate that patients with KD carry a higher risk for peripheral arteriopathy. This naturally does not relate to KD’s coronary artery-related vasculopathy. Invasive high-resolution investigation of the coronary arteries in KD patients, such as optical coherence tomography (OCT), indicates an altered vascular-wall texture similar to intimal hyperplasia, partial disappearance of media and fibrosis, seen at both aneurysmal sites and angiographically normal segments [30]. Yet, little is known about coronary vascular wall texture in KD children, never having had evidence of coronary artery involvement, since invasive cardiac catheterization is usually not part of routine follow-up investigation.

### 4.2. Strengths and Limitations

In this study, we investigated the cIMR in patients with a history of KD in comparison to healthy controls. By recruiting groups whose anthropometric and laboratory data are similar—that is, our two groups did not differ significantly in sex, age, weight, height and laboratory data—we restricted influencing factors to a minimum. However, our data should be interpreted in the light of the low number of study participants. We had only a few patients with coronary artery aneurysms, thus weakening our statistical subgroup analysis. Furthermore, the broad range of the time since the KD diagnosis must also be accounted for. We were unable to compare our results within a clinical context as there are no cIMR normative values applying to children < 8 years, or even to adults, available.

We assessed the systolic and diastolic cIMR, which differed significantly, indicating that, in future cIMR studies, clear reference to heart cycle-specific values may be useful. However, the differences are small and should be confirmed by studies including a larger sample size.

## 5. Conclusions

In this study, cIMR did not differ in either patients with a KD history or in those with KD and a severe cardiovascular phenotype, such as persistent CAA. Therefore, our results do not support the hypothesis of a generally higher long-term cardiovascular risk in patients with KD. Nevertheless, more investigations are needed with larger study populations that investigate other known vascular parameters in association with cIMR. Nevertheless, CIMR seems to be a promising parameter in the long-term assessment of atherogenesis.

## Figures and Tables

**Figure 1 jcm-11-01177-f001:**
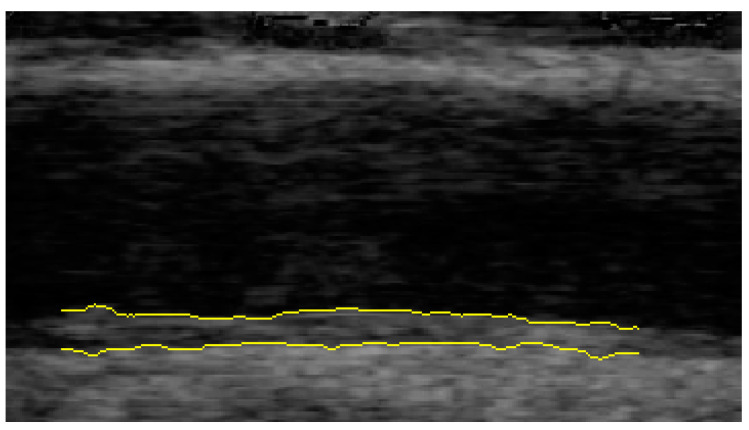
Automated identification of intima–media and media–adventitia border of the carotid artery using the software DYARA.

**Figure 2 jcm-11-01177-f002:**
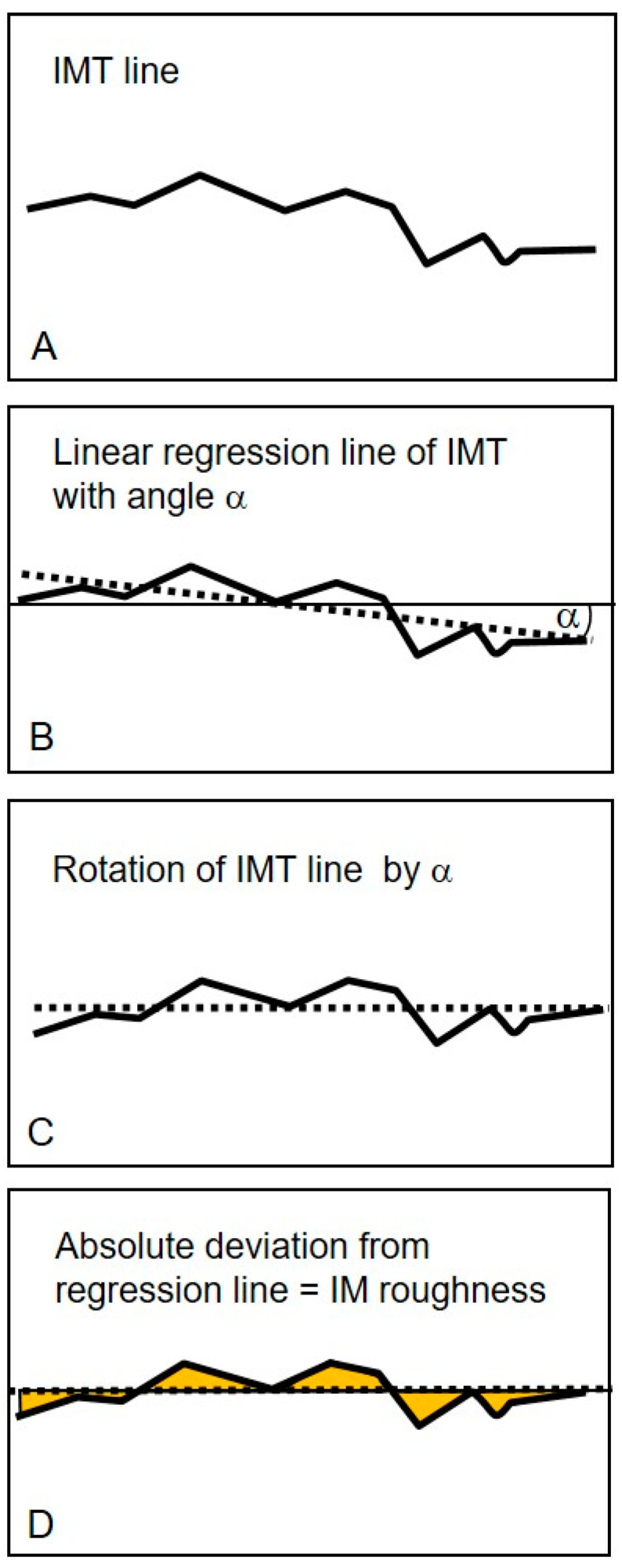
Schematic illustration of the calculation of the carotid intima media roughness (cIMR) adapted from Schmidt-Trucksäss et al. *Atherosclerosis* (2003) [11]. (**A**): carotid intima media thickness (cIMT) line calculated as the difference between all intima–lumen and media–adventitia measurement points. (**B**): shows the linear regression line of all cIMT measurement points, with α representing the angle between the regression line and the horizontal. (**C**): Rotation of regression line to horizontal (**D**): The yellow colored areas indicate the profile deviation of the cIMT from the regression line. The arithmetic mean of this deviation is equivalent to the cIMR.

**Figure 3 jcm-11-01177-f003:**
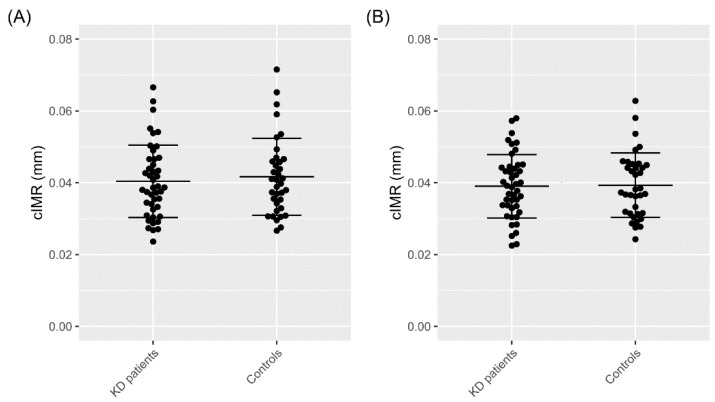
cIMR measurements expressed as mean ± standard deviation. (**A**) End-diastolic cIMR in patients with Kawasaki disease (KD) (0.040 ± 0.010 mm, N = 44) vs. healthy controls (0.042 ± 0.011 mm, N = 36), *p*-value = 0.590; (**B**) end-systolic cIMR KD patients (0.039 ± 0.009 mm, N = 44) vs. healthy controls (0.039 ± 0.009 mm, N = 36), *p*-value = 0.886. Differences between groups were analyzed via Welch′s unequal variances *t*-test.

**Table 1 jcm-11-01177-t001:** Baseline characteristics of controls and patients with KD.

	*n*	KD Patients (*n* = 44)	*n*	Controls (*n* = 36)	*p*-Value *
Female, *n* (%)	44	15 (34.1)	36	14 (38.9)	0.657
Age (years)	44	13.4 (7.5)	36	12.1 (5.3)	0.372
Height (cm)	44	149.3 (24.2)	36	148.4 (22.5)	0.864
Weight (kg)	44	43.5 (21.3)	36	42.2 (18.6)	0.763
BMI (kg/m^2^)	44	18.2 (3.7)	36	18.1 (3.1)	0.834
Blood pressure					
SBP (mmHg)	44	117.3 (13.6)	36	113.4 (8.2)	0.117
DBP (mmHg)	44	70.9 (10.0)	36	68.2 (8.9)	0.207
MAP (mmHg)	44	92.1 (10.5)	36	89 (7.7)	0.128
HR (1/min)	44	88.5 (12.4)	36	82.9 (12.6)	0.053
Laboratory data					
Total cholesterol (mg/dL)	38	163.6 (31.5)	24	171.2 (30.7)	0.356
Triglycerides (mg/dL)	38	112.5 (65.3)	24	88.5 (35.4)	0.067
LDL (mg/dL)	26	104.7 (25.5)	24	104 (26.2)	0.925
VLDL (mg/dL)	20	16.6 (6.4)	24	13.9 (6.5)	0.176
HDL (mg/dL)	26	50.1 (13.0)	24	53.9 (8.7)	0.224
CAA Status					
Group A		27 (69.2)			
Group B		6 (15.4)			
Group C		6 (15.4)			

BMI, body mass index; CAA, coronary artery aneurysm; DBP, diastolic blood pressure; HDL, high-density lipoprotein; HR, heart rate; KD, Kawasaki disease; LDL, low-density lipoprotein; MAP, mean arterial pressure; VLDL, very low-density lipoprotein. Data are expressed as the mean (standard deviation) unless otherwise specified. * Differences between groups were analyzed with the χ2 test for sex distribution and with Welch′s unequal variances *t*-test for all other variables. Group A: never had CAA, Group B: regressed CAA, Group C: persisting CAA; missing data of 5 patients.

**Table 2 jcm-11-01177-t002:** Association between KD and cIMR.

	cIMR End-Diastolic (mm)	*p*-Value	cIMR End-Systolic (mm)	*p*-Value
Kawasaki disease (Yes vs. No)	−0.128	0.584	−0.044	0.851

cIMR, carotid intima–media roughness. Standardized β coefficients and *p*-values for two linear regression models (each with sample size N = 80) with Kawasaki disease (yes vs. no) encoded as a dummy predictor variable and cIMR end-diastolic (mm) resp. cIMR end-systolic (mm) as the dependent variable. Adjustment for the following covariates was performed: sex, age, height and weight. All continuous variables, i.e., cIMR, age, height and weight were standardized into z-scores, resulting in the reported standardized β coefficients.

**Table 3 jcm-11-01177-t003:** Acute phase parameters in KD patients.

	*n*	
Aneurysms (cases, %)	39	12 (30.8%)
Therapy refractory (>1 times IVIG) (cases, %)	37	14 (37.8%)
Duration of fever (days)	33	9.70 (5.3)
Age at disease (months)	41	4.4 (3.2)
BMI (kg/m^2^)	30	15.3 (1.4)
CRP (mg/L)	35	115.4 (92.0)
Thrombocytes (g/L)	32	534.9 (175.6)
Hb (g/dL)	34	10.2 (1.9)
Leukocytes (g/L)	35	19.1 (10.9)

BMI, body mass index; CRP, C-reactive protein; Hb, hemoglobin; IVIG, intravenous immunoglobulins; KD, Kawasaki Disease. Data are expressed as the mean (standard deviation) unless otherwise specified.

**Table 4 jcm-11-01177-t004:** Association between coronary artery status and cIMR in KD patients.

	cIMR End-Diastolic (mm)	*p*-Value	cIMR End-Systolic (mm)	*p*-Value
Group A (vs. healthy)	−0.140	0.599	−0.143	0.583
Group B (vs. healthy)	−0.403	0.391	0.293	0.524
Group C (vs. healthy)	0.006	0.990	0.280	0.554
Group B + C (vs. A)	−0.056	0.888	0.543	0.150

cIMR, carotid intima-media roughness. Group A: No CAA; Group B: CAA at acute phase; Group C: persistent CAA. Standardized β coefficients and *p*-values from four different linear regression models, resulting from two different dependent variables (cIMR end-diastolic and cIMR end-systolic) and two types of predictors. For the first predictor type (sample size N = 75), groups A, B and C were each compared to the healthy control group, by encoding groups A, B, C vs. healthy as dummy predictor variables with the healthy group as baseline. The comparisons of A, B and C to the healthy control group were performed simultaneously in a single model. For the second predictor type (sample size N = 39), the group composed of both B and C was compared to group A, by encoding group B + C vs. group A as a dummy predictor variable with group A as the baseline. Healthy controls were not included in the latter type of model. Adjustment for the following covariates was performed: sex, age, height and weight. All continuous variables, i.e., cIMR, age, height and weight were standardized into z-scores, resulting in the reported standardized β coefficients.

## Data Availability

Data available on request due to restrictions eg privacy or ethical.

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
