# Peer review of "KD Diagnosis Does Not Increase Cardiovascular Risk in Children According to Dynamic Intima–Media Roughness Measurements"

_jcm, 2022, doi:10.3390/jcm11051177_

Round 1

Reviewer 1 Report

  1. The article name “Dynamic Intima media roughness. Possible long-term vascular alterations in patients with Kawasaki disease?” is interesting and useful article for vascular complication related to KD.
  2. The study design and aim of the study are scientifically sounded and well analyzed despite the sample size is small.
  3. Data on the Findings of cIMR by Dynamic Artery Analysis (DYARA) software were well interpreted and tabulated.

4.In this study, the researchers are wise to analyses cIMR  as  a  potential  surrogate  marker  for  long-term  vascular  anomalies  in  children  with  a  history of  KD.

  1. Overall, a good article on cIMR with KD for readers and scientific communities.

Author Response

Dear Reviewer,

we thank you for the encouraging words, the well thought reviews and recognition of our efforts.

We look forward to hearing from you at your convenience.

Sincerely,

André Jakob (Corresponding author)

Reviewer 2 Report

It is a surprisingly dissapointing manuscript which is treating about a good scientific idea and interesting medical observation but with a wrong and misleading description. I do understand that the results must have failed to meet the authors' hypothesis but they could have drawn better conclusions from this situation. Massive parts of the manuscript need to be rewritten.

Here are the major flaws that I have found:

  1. The title does not fit the findings of the authors. According to the results, intima media roughness cannot be considered as the marker of long-term vascular alterations. In my view the strongest conclusion of the article is presented in the lines 328-329:
    "our results do not support the indication of a generally higher long-term cardiovascular risk in patients with KD".
    I suggest changing the general idea of the article and rendering it to the vision under the working title "KD diagnosis does not increase cardiovascular risk in children according to intima media roughness measurements".
  2. The authors are trying to convince potential readers that "the end-diastolic cIMR was significantly rougher than end-systolic values" (lines 29-31 in the Abstract). The marginally significant finding (p=0.044) based on incomprehensible group (combining both KD patients and healthy controls) which underlines the difference of 0.002mm does not deserve to be considered as the main result of this interesting study. It is also worth interjecting here that the resolution and the sensitivity of USG measurement has not been discussed by the authors. I recommend abandoning this track and continuation of my aforementioned suggestion.
  3. Dividing an already small group of patients into subgroups is not substantially defined and is not included in the general characteristics of the study group (Table 1). The reader is confused about the general objective and results of this process. 
  4. There are numerous language errors including misspellings (e.g. comtrolls line 20), inconsistent punctuation (e.g. dots line 22 vs. coma line 23) and informal short forms (e.g. doesn't line 69). The next manuscript requires professional English editing before submission.

Reviewer 3 Report

I enjoyed reading the manuscript "Dynamic Intima media roughness. Possible long-term vascular alterations in patients with Kawasaki disease?". The manuscript is well written and the methods seem sound. Although the authors hypothesis was clearly not confirmed, the results are clinically important; Indeed many questions remain about the long term fate of vasculature, especially the coronary vasculature in patients with KD treated with IVIG. I have one major issue with the methodology and hypothesis used by the authors. They evaluated results of measurements that have been proven to be correlated with atherosclerosis and hence coronary disease in adults. But coronary problems in adults are closely related to atherosclerosis and its contributing factors: ageing, smoking and elevated cholesterol. And although KD patients suffer from coronary disease during and after the acute phase of the disease, peripheral vascular disease is less common in these patients and coronary disease caused by KD is probably not associated to peripheral vascular disease as is the case in general atherosclerosis. To me it is no surprise that there is no relationship with a marker of general vascular health/disease and coronary damage in KD patients. Is there any similar measurement of the coronary vessels specifically that could be used as a more specific marker in KD patients, reflecting their coronary health and maybe long term coronary risk factors? Please could you comment?

Round 2

Reviewer 2 Report

Thank you for implementation of my suggestions. I do believe that they have strongly elevated the quality of the manuscript. My only remaining concern is related to end-diastolic cIMR vs. end-systolic values (this issue has been already discussed in the previous report). The authors have rendered the main text to live up to my expectations, but this marginal finding (p=0.044) of the difference of 0.002mm remained in the Abstract (lines 29-31). I suggest rephrasing the sentence into "However, according to our marginally significant findings (p=0.044) we postulate that the end-diastolic cIMR may be rougher than the end-systolic values in KD patients" (without means and 95% CI in brackets).

Author Response

Dear Reviewer,

we thank you very much for appreciating our efforts to meet your expectations, we do agree that those changes elevated the quality of our manuscript. Furthermore we changed the sentence in line 29-31 according to you suggestion. If any further concerns arise, please let us know.

We look forward to hearing from you at your convenience.

Sincerely,

André Jakob (Corresponding author)